

# Conflict-induced ship traffic disruptions constrain cloud sensitivity to stricter marine pollution regulations

Michael S. Diamond[1] and Lili F. Boss[1]

[1]Department of Earth, Ocean, & Atmospheric Science, Florida State University, Tallahassee, FL 32306, USA

*Correspondence to*: Michael S. Diamond (msdiamond@fsu.edu)

**Abstract.** Starting in November 2023, the Houthi militia occupying northeastern Yemen has attacked ships passing through the Bab al-Mandab Strait, a chokepoint on the Europe-Asia route via the Suez Canal. Cargo ship traffic through the Red Sea has since plummeted, with ships instead taking the longer route around the Cape of Good Hope. The increase in traffic in the southeastern Atlantic Ocean is readily apparent in satellite retrievals of nitrogen dioxide. Within the stratocumulus deck

covering much of the southeastern Atlantic, a previously detectible cloud microphysical perturbation due to ship pollution had largely disappeared following the International Maritime Organization's sulfur-limiting regulations in 2020 but returns during 2024 due to the increase in ship traffic despite the lower cloud brightening efficacy per ship. Because nitrogen dioxide pollution per unit of fuel oil burned is not affected by switching to low-sulfur fuel, quantifying the ratio of shipping-enhanced cloud droplet number and nitrogen dioxide concentrations before and after the fuel sulfur limits went into effect provides a constraint

on the cloud changes from the regulations. We find that the ~80% reduction in sulfur emissions leads to a ~66% reduction in the increase in cloud droplet number concentration per unit marine fuel oil burned.

## 1 Introduction and approach

How clouds respond to changes in airborne particles (aerosols) is the single largest source of uncertainty in quantifying the anthropogenic perturbation to Earth's radiation budget (Bellouin et al., 2020; Forster et al., 2021). Ship tracks, curvilinear

streaks of brighter clouds trailing ship smokestacks (Conover, 1966), are the quintessential example of a "natural experiment" in aerosol-cloud interactions (Christensen et al., 2022) as they provide compelling causal evidence for aerosol effects on cloud properties with a very limited role for meteorological confounding over the order 10 km width of a typical track (Durkee et al., 2000). Ship-emitted aerosol particles that are large and/or hygroscopic enough to serve as cloud condensation nuclei (CCN) increase cloud droplet number concentration ($N_d$); for a given amount of cloud water, this results in brighter clouds that can

reflect more sunlight back to space and produce a cooling effect (Twomey, 1977). Macrophysical adjustments to this change in cloud microphysics can enhance the cooling by reducing drizzle, thus increasing cloud coverage (Albrecht, 1989; Radke et al., 1989; Yuan et al., 2023), or counteract it by encouraging greater turbulent entrainment of dry above-cloud air into the marine boundary layer, thus reducing cloud thickness (Chen et al., 2012; Coakley and Walsh, 2002; Toll et al., 2019).



Starting in January 2020, marine fuel sulfur content restrictions from the International Maritime Organization (IMO 2020) have resulted in an approximately 80% reduction in sulfur emissions (Diamond, 2023), decreasing the size and hygroscopicity of the remaining aerosol particles and thus their ability to act as CCN (Petzold et al., 2010). Decreases in the detection of ship tracks over open oceans (Watson-Parris et al., 2022; Yuan et al., 2022) and the near disappearance of detectible cloud microphysical perturbations within an isolated shipping corridor in the southeast Atlantic (Benas et al., 2025; Diamond, 2023)

have been observed, but the estimated magnitude of the IMO 2020 effect ranges from ~15-75% depending on methodology (Gettelman et al., 2024).

An opportunity to better constrain the effect of the IMO 2020 has arisen from an unlikely source: substantial changes in ship traffic patterns following Houthi militia attacks on merchant vessels (Raydan and Nadimi, 2025) transiting through the Bab

al-Mandab Strait connecting the Indian Ocean and Mediterranean Sea via the Suez Canal. Analysis of satellite-derived (van Geffen et al., 2022; Veefkind et al., 2012) tropospheric columns of nitrogen dioxide ($NO_2$) has revealed clear decreases in ship traffic in the Red Sea and increases around the Cape of Good Hope following the commencement of hostilities in November 2023 (Pseftogkas et al., 2024), consistent with rerouting cargo ships around Africa to avoid the Suez Canal. Because nitrogen oxide ($NO_x$) production is dominated by high temperatures during combustion and is largely unaffected by fuel content or the

installation of sulfur scrubbers, if shipping $NO_2$ perturbations and cloud microphysical changes in the southeastern Atlantic stratocumulus deck could be quantified both before IMO 2020 and after the increase in traffic due to the Red Sea crisis, their ratio would provide an estimate of the cloud altering effects per unit marine fuel burned pre- and post-regulation.

## 2 Results

Figure 1 contrasts column $NO_2$ concentrations from September-October 2023 and 2024 observed by the TROPOspheric

Monitoring Instrument (TROPOMI; van Geffen et al., 2022) over the Red Sea (a, b) and southeast Atlantic (c, d). Decreased ship activity in the Red Sea and the related increase in the southeast Atlantic from 2023 to 2024 is readily visible; shipping corridors are difficult to discern in the southeast Atlantic in 2023 but are clear after the Houthi attacks began (and vice versa for the Red Sea). Because $NO_2$ is difficult to retrieve in the overcast conditions characteristic of the stratocumulus deck (red box in Fig. 1d), the $NO_2$ perturbation from shipping is quantified including less cloudy regions to the north (blue box in Fig.

1d; see Methods in Appendix A).

The 2024 $NO_2$ perturbation clearly stands out as greater than prior years (Fig. 2a). Essentially no change is seen during the peak of the COVID-19 pandemic in 2020, in line with previous work suggesting minimal ship traffic disruptions in the region (March et al., 2021). Consistent with prior results for the shipping corridor cutting through the southeastern Atlantic

stratocumulus deck (Benas et al., 2025; Diamond, 2023; Diamond et al., 2020; Hu et al., 2021), the relative $N_d$ perturbation ($\Delta \ln N_d$) from the Visible Infrared Imaging Radiometer Suite (VIIRS) on NOAA-20 (Fig 2b; see Methods in Appendix A) is





5-10% before 2020 and indistinguishable from zero afterward, until rebounding to ~6% (95% confidence interval: 3-9%) in 2024.

Taking the ratio of the relative $NO_2$ and $N_d$ perturbations (Fig. 2c) reveals a marked decrease in the cloud-altering efficiency of marine fuel oils after the IMO 2020 regulations went into effect, including in 2024 when the $N_d$ perturbation is sizeable (Fig. 2b). Because the 2018 $NO_2$ values have a particularly low tail and $N_d$ values have a high tail, the ratio for that year is poorly constrained in the mean and on the upper end. As the post-2020 but pre-Red Sea crisis $N_d$ perturbations are indistinguishable from zero at 95% confidence, their ratios are also poorly constrained. We thus take the 2019 ratio of 1.4 (0.7-
3.0) as representative of the pre-IMO 2020 period and contrast with the 2024 post-IMO 2020 value of 0.5 (0.2-0.8). Comparing the 2019 and 2024 sensitivities of cloud properties to unit fuel burnt shows a 66% (25-88%) reduction in the ability of ship emissions to influence cloud microphysics following IMO 2020.

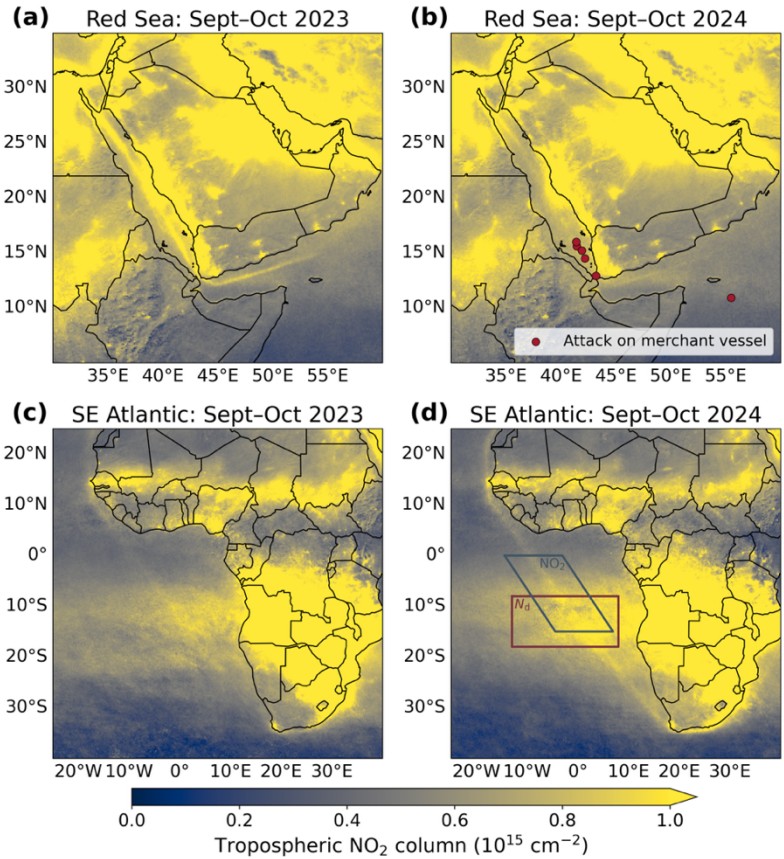

**Figure 1. TROPOMI $NO_2$ values in the Red Sea (a, b) and southeast Atlantic (c, d) before (a, c) and after (b, d) the Red Sea crisis.**
**Red markers (b) represent locations of attacks on merchant vessels during September–October 2024. Boxes in (d) indicate the analysis domains for the $NO_2$ (blue) and $N_d$ (red) data.**



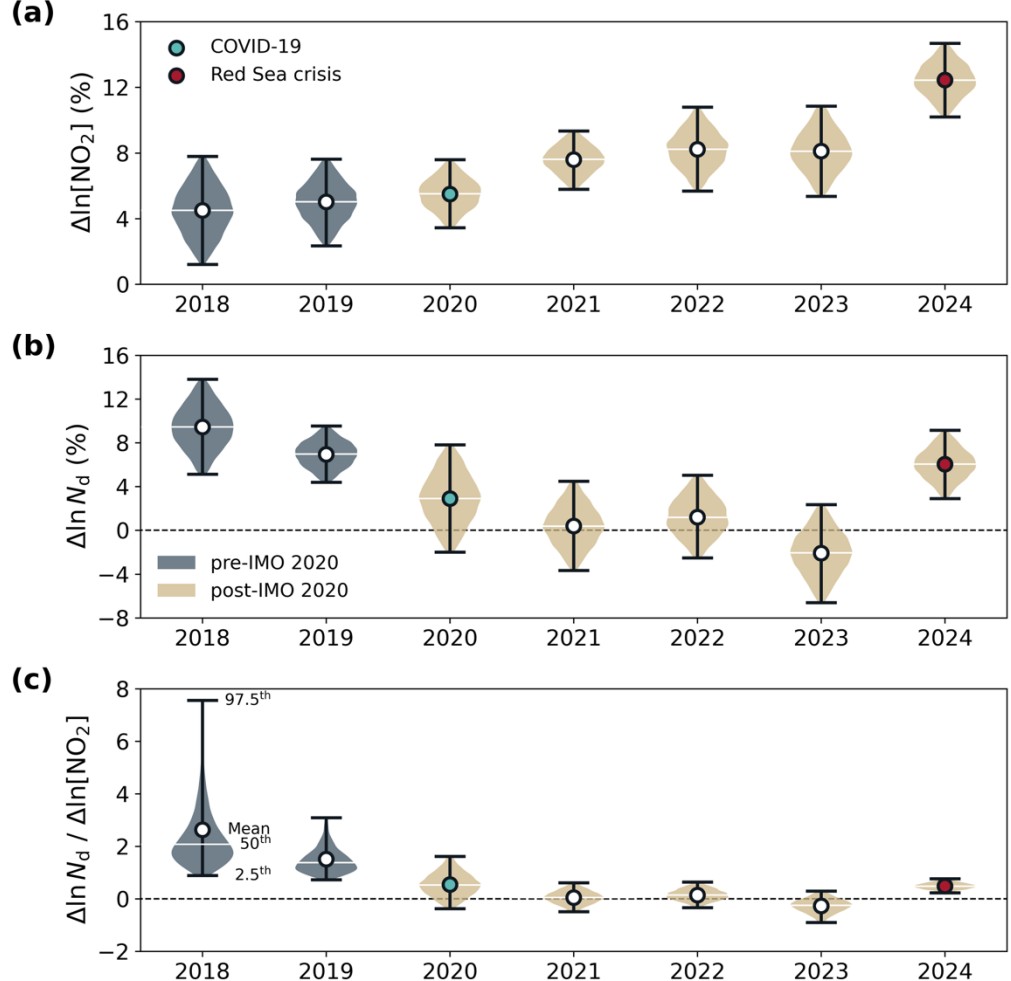

**Figure 2. Relative shipping perturbations in (a) NO₂, (b) $N_d$, and (c) their ratios. Violins represent 95% confidence, white lines median, and markers mean values for each distribution.**

## 3 Discussion

### 3.1 Comparison with detectable ship track methods

Combining the 66% reduction in the $N_d$-NO₂ ratios from Fig. 2c with the ~80% reduction in sulfur gives a relative sensitivity of 80% (30-110%), which is more linear in terms of cloud changes per sulfur decline than reported estimates from ship track detections. Over the same region (8-18°S, 13°W-8°E) and season (September–October), a typical year from the 2003-2019 period had 116 ± 61 detected ship tracks in the dataset of Yuan et al. (2022). 59 were detected in 2020, implying a decline in the sensitivity of clouds to ship emissions after IMO 2020 of 49% ± 27%, or a 61% ± 34% relative sensitivity accounting for the sulfur change. Similar results are reported for the southeastern Atlantic in Watson-Parris et al. (2022). There are reasons



to believe that estimates using a binary detection threshold metric should systematically underestimate the linearity of the cloud response to shipping emissions, however. Indeed, a perfect detection algorithm would in theory show no decline in ship tracks post-IMO 2020 so long as some CCN remain, even if the cloud perturbations within the track were greatly diminished. Estimates based on cloud properties may therefore be more representative of the radiative forcing implications of the IMO 2020 regulations, and thus possible temperature changes (England et al., 2025; Jordan and Henry, 2024; Quaglia and Visioni, 2024; Raghuraman et al., 2024; Watson-Parris et al., 2025), than those based on ship track detectability. $N_d$ perturbations also need to be interpreted with caution in extrapolating to effective radiative forcing, as the ultimate climate effects are highly sensitive to adjustments in cloud amount if the microphysical changes affect precipitation and turbulent mixing.

## 3.2 Robustness of single-year estimates

A major limitation of the present analysis is the reliance on estimates derived from single years of two-month data. Even for the clear increase in $N_d$ observable over the shipping corridor, there is substantial noise in single-year estimates of the shipping effect (e.g., Fig. 6 in Diamond et al., 2020). To test how unusual it would be to observe something like the decrease in $N_d$ perturbation in all years after 2019 and before 2024, we repeat the analysis above for VIIRS on NOAA-2020 (available since 2018) with values from the MODerate resolution Imaging Spectrometer (MODIS) on Aqua (available since 2002, with increasing orbital drift since 2022; see Methods in Appendix A). Figure 3 shows how each individual year's data compares to the pre-2020 climatology, the 2020-2023 period, and 2024. Using a Monte-Carlo approach, we randomly draw 100,000 samples, with replacement, of four-year sets from the population of single-year (September-October) Aqua/MODIS data from 2003-2019 and test how frequently the mean $N_d$ for all four years is less than half the climatological 2003-2019 value (as was true for 2020-2023). Such a situation occurs in less than 0.02% of cases. It is thus exceedingly unlikely that the pattern observed in the single-year MODIS or VIIRS data since IMO 2020 went into effect is due to chance.

## 4 Conclusions

Changes in ship traffic stemming from international conflict around the Red Sea came just a few years after ship emissions had already changed dramatically due to sulfur pollution regulations; this unique "experiment-within-an-experiment" setup allows us to better quantify how cloud responses to shipping have changed following IMO 2020. By comparing the shipping perturbation to $NO_2$, which is not strongly affected by fuel composition, to the shipping effect on CCN (as indicated by $N_d$), which is very sensitive to the fuel composition, we derive a measure of the cloud-altering properties of a given quantity of fuel burned. The spike in shipping activity in the southeast Atlantic in 2024 due to disruption in Suez Canal traffic results in raw $N_d$ perturbations rebounding from being indistinguishable from zero post-IMO to being nearly as large as the pre-2020 values; normalizing by the anomalously large $NO_2$ perturbation, however, reveals a 66% (25-88%) reduction in the cloud-altering ability of the post-2020 fuel, at least in stratocumulus conditions. If the constraint on cloud microphysical sensitivity to





shipping emissions established here could be extrapolated globally, it may provide an emergent constraint on historical radiative forcing from aerosol-cloud interactions from global climate models.

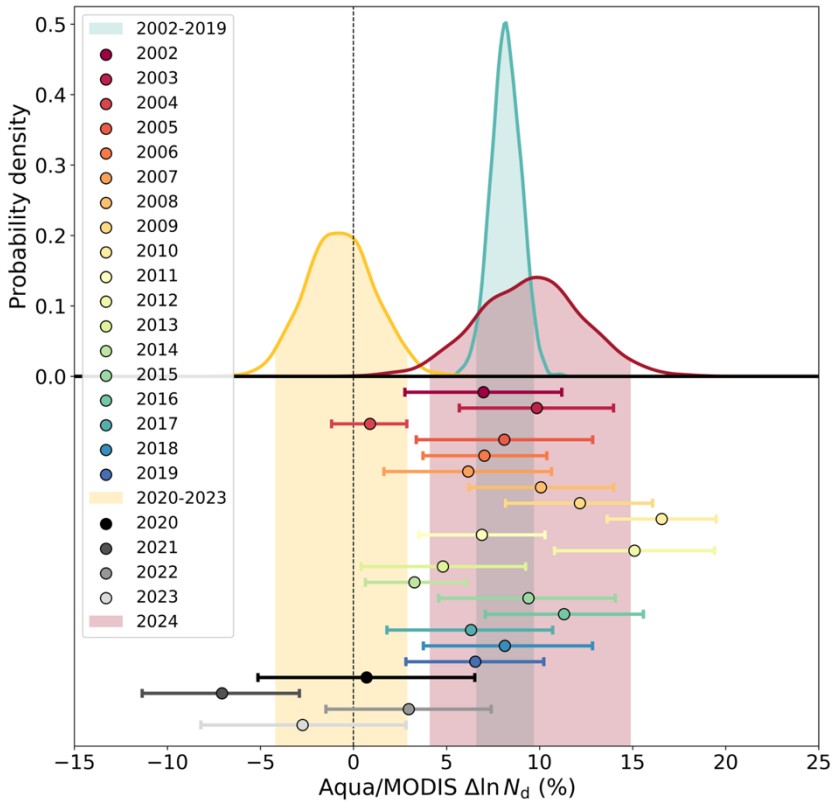


**Figure 3. Probability densities (via Gaussian kernel density estimation) of relative $N_d$ perturbations for the pre-2020 September-October climatology (blue curve), 2020-2023 period (yellow curve), and 2024 (red curve), with 95% confidence intervals shown as transparent shading, and individual year mean estimates (colored markers) and their 95% confidence intervals (error bars) from Aqua/MODIS.**

**Appendix A: Methods**

NO$_2$ data (TM5-MP-DOMINO v2.8.0) is from the TROPOspheric Monitoring Instrument (TROPOMI) on Sentinel 5-P (Copernicus Sentinel-5P, 2021; van Geffen et al., 2022). The shipping corridor contribution to column NO$_2$ is calculated by first finding the longitude with the greatest shipping NO$_x$ emissions from the Emissions Database for Global Atmospheric Research (EDGAR; Crippa et al., 2020) for each latitude in the domain 13.5°W-3.5°E, 15°S-0°, compositing the data 5° east

and west of these points, fitting a line connecting the data between 2-5° to the east and west, and subtracting the regressed



values from the observed values within 0.25° east and west of the central point. Errors are quantified by propagating the sampling uncertainty in the observed mean value and the root mean square error of the regression.

$N_d$ is calculated (Grosvenor et al., 2018) from the Visible Infrared Imaging Radiometer Suite (VIIRS) on NOAA-20 and the MODerate resolution Imagaing Spectrometer (MODIS) on Aqua (Platnick et al., 2017) using joint histograms of liquid cloud droplet effective radius and optical thickness following Painemal and Zuidema (2011). The shipping corridor contribution to $N_d$ is estimated by generating a counterfactual field of non-shipping-influenced values following the universal kriging methodology of Diamond et al. (2020). The definition of potentially shipping-affected pixels follows Diamond (2023) but is expanded 2° to the west; the "core" corridor effect is quantified as the average difference between 5000 simulated plausible

counterfactual values and the observed value at the grid points of the greatest shipping $SO_2$ emissions (Crippa et al., 2020) and 2° to the west at each latitude (18-8°S). Sea surface temperature from the Extended Reconstructed Sea Surface Temperature, Version 5 dataset (Huang et al., 2017a) is used as the cloud controlling factor in fitting the mean function.

Unless otherwise specified, reported central values represent medians with 95% confidence intervals in parentheses.

**Open science:**

TROPOMI $NO_2$ data were downloaded from the Tropospheric Emission Monitoring Internet Service at https://www.temis.nl/airpollution/no2.php (last access: 30 July 2025) are also available at https://doi.org/10.5270/S5P-9bnp8q8 (Copernicus Sentinel-5P, 2021). NOAA-20/VIIRS and Aqua/MODIS data are available from the NASA Level-1 and Atmosphere Archive & Distribution System (LAADS) Distributed Active Archive Center (DAAC) at

https://doi.org/10.5067/VIIRS/CLDPROP_M3_VIIRS_NOAA20.011 (NASA VIIRS Atmosphere SIPS) and https://doi.org/10.5067/MODIS/CLDPROP_M3_MODIS_Aqua.011 (NASA LAADS DAAC), respectively. The Emissions Database for Global Atmospheric Research is available from the European Commission, Joint Research Centre at https://doi.org/10.2904/JRC_DATASET_EDGAR (European Commission Joint Research Centre, 2018). NOAA Extended Reconstructed SST V5 data are available at https://doi.org/10.7289/V5T72FNM (Huang et al., 2017b). Locations of attacks

on merchant vessels are from Raydan and Nadimi (2025), available at https://www.washingtoninstitute.org/policy-analysis/tracking-maritime-attacks-middle-east-2019 (last access: 30 July 2025). Ship track locations from Yuan et al. (2022) are available from Harvard Dataverse at https://doi.org/10.7910/DVN/JII4DN (Song, 2022). The following R and Python libraries were integral to the analysis: cartopy (Met Office, 2010-2015), geoR (Ribeiro and Diggle, 2018), matplotlib (Hunter, 2007), numpy (Harris et al., 2020), scipy (Virtanen et al., 2020), and xarray (Hoyer and Hamman, 2017). Code and processed

data used in this analysis are available at https://doi.org/10.5281/zenodo.16637711 (Diamond, 2025) and https://doi.org/10.5281/zenodo.15738910 (Diamond and Boss, 2025), respectively.



**Acknowledgments**

MSD and LFB were supported by NOAA's Climate Program Office Earth's Radiation Budget (ERB), Atmospheric Chemistry, Carbon Cycle, & Climate (AC4), and Climate Variability & Predictability (CVP) Programs, Grant NA23OAR4310297.

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
