# Peer review of "Conflict-induced ship traffic disruptions constrain cloud sensitivity to stricter marine pollution regulations"

_EGUsphere, 2025_

## Author Comment (AC1)

We thank both reviewers for their thoughtful comments and have improved the manuscript following their helpful questions and suggestions. Please find a point-by-point response below, with reviewer comments in italics and author response in blue. -Michael Diamond

**Anonymous Reviewer 1**

The paper uses the change in cloud drop number concentration ratioed to the change in NO2 to assess how the IMO's implementation of sulfur-limiting regulations in 2020 affected cloud brightening due to ship tracks. The SE Atlantic is the area of focus due to recent increases in ship traffic as ships transit around the Cape of Good Hope rather than through the Suez Canal to avoid Houthi militia attacks. Comparing years 2019 and 2024, a large decrease in the ability of ship emissions to impact Nd is found. The paper is concise and well-written. The figures are of high quality and effectively display the results reported in the paper. I only have a few minor comments.

We thank the anonymous reviewer for their positive appraisal and constructive suggestions to improve clarity.

Line 35: "the estimated magnitude of the IMO 2020 effect ranges from ~15-75%". Is this based on the reduction in detectible ship tracks? Please clarify.

This is based on a combination of different studies and methodologies including ship track detection, air-mass tracking of ship locations, and analysis of the SE shipping corridor, as summarized in the assessment of Gettelman et al. (2024). The text has been revised to clarify this point (Track Changes lines 37-41).

Lines 67 - 68: "Because the 2018 NO2 values have a particularly low tail and Nd values have a high tail..." I could be interpreting this statement incorrectly but the 2018 high tail of the Nd values is not evident in Figure 2b.

Although both underlying NO2 and Nd distributions are approximately normal, the important point to make here is that the lower end of the 2018 NO2 values are particularly low (only year with substantial weight below 2.5%) and the upper end of the 2018 Nd values are particularly high (only year with substantial weight above 10%). The text has been revised to clarify this point (Track Changes lines 74-75).

Line 88: Does the "binary detection threshold metric" refer to relying on only two years to assess the cloud response to the decrease in ship sulfur emissions?

Apologies for the confusion — this refers to a detection metric which is binary in that it only has two options ("ship track detected" versus "no ship track detected"). The importance here is that a binary metric like this cannot provide useful information like "a ship track is detected but it's 67% weaker than before". Worse, the better the detection algorithm is, the poorer it will be at providing information about weakening but still-existent tracks. The text has been revised to clarify this point (Track Changes lines 97-103).

**Jay Mace (Reviewer 2)**

The study by Diamond and Boss examining the change in cloud droplet number (Nd) in the SE Atlantic region following the increase in shipping there during 2024 relative to the years since the fuel change and relative to the pre 2020 results shows that Nd increased in proportion to the increase in traffic. This allows the authors to quantify the efficiency with which Nd is influenced by shipping exhaust from before and after the 2020 change in sulfur content. Overall, I find the paper to be straightforward, concise, and compelling. I suggest only minor revisions.

We thank Jay Mace for his positive appraisal and constructive suggestions to more rigorously test for meteorological influences that could complicate single-year analyses and to improve the clarity of the manuscript.

The authors acknowledge the fact that examining a single year (2024) for the increase in traffic in their study region is somewhat fraught. They make a compelling and convincing argument that their findings are significant. However, I think they need to at least examine the large-scale atmosphere during 2024 to see if the months they consider are typical or perhaps anomalous relative to other years. The reason I think this is necessary is due to the role of drizzle in modulating Nd. While their data do not constrain drizzle occurrence or rate, were, perhaps, the marine inversion different from other years, drizzle may be more or less common, etc. While I think their results will hold up against this examination, it would be at least useful to examine this and report upon it

Thanks for the excellent suggestion, and we agree that accounting for the meteorological state, particularly in terms of drizzle, would increase the rigor of using single-year estimates.

See new Figure 4 and Track Changes lines 127-152 (Section 3.2) for a discussion of our analysis of CERES SSF NOAA-20 cloud fraction, cloud effective radius, aerosol optical depth, SST, EIS, and wind speed data. In brief, no concerning outliers are identified.

Lines 125-130 (first paragraph in the methods): It took me a bit to digest this and realize that the delta values in figure 2 were relative to the regression they discuss (do I have that right?). I think an illustrative figure would be helpful here to show their method. I think the approach is sound, but understanding their text would benefit greatly from a figure.

Thanks for another excellent suggestion (and yes, your understanding is correct). See new Figure A1 for an illustration of the approach using the September-October 2024 data.

Caption to Figure 2: Note that the region displayed is from SE Atlantic. This is somewhat obvious but it would still be helpful to casual reader who might just scan the figures.

Great point; clarification has been added to the caption of Figure 2.

With compliments to the authors, Jay Mace

---

## Author Comment (AC2)

[revised manuscript text omitted]

**90 3 Discussion**

**3.1 Comparison with detectable ship track methods**

Combining the 66% reduction in the  $N_d$ -NO2 ratios from Fig. 2c with the ~80% reduction in sulfur gives a relative sensitivity of 80% (30-110%), which is more linear in terms of cloud changes per sulfur decline than reported estimates from ship track detections. Over the same region (8-18°S, 13°W-8°E) and season (September–October), a typical year from the 2003-2019 period had 116  $\pm$  61 detected ship tracks in the dataset of Yuan et al. (2022). 59 were detected in 2020, implying a decline in the sensitivity of clouds to ship emissions after IMO 2020 of 49%  $\pm$  27%, or a 61%  $\pm$  34% relative sensitivity accounting for

the sulfur change. Similar results are reported for the southeastern Atlantic in Watson-Parris et al. (2022), with global values even lower ( $\sim$ 25% reduction in tracks, or  $\sim$ 30% relative susceptibility). There are reasons to believe that estimates using a binary detection threshold metric (i.e., "ship track detected" versus "no ship track detected") should systematically underestimate the linearity of the cloud response to shipping emissions, however. Indeed, a perfect detection algorithm would in theory show no decline in ship tracks post-IMO 2020 so long as some CCN remain, even if the cloud perturbations within the detected ship track were greatly diminished. Estimates based on cloud properties may therefore be more representative of the radiative forcing implications of the IMO 2020 regulations, and thus possible temperature changes (England et al., 2025; Jordan and Henry, 2024; Quaglia and Visioni, 2024; Raghuraman et al., 2024; Watson-Parris et al., 2025), than those based on ship track detectability.  $N_d$  perturbations also need to be interpreted with caution in extrapolating to effective radiative forcing, as the ultimate climate effects are highly sensitive to adjustments in cloud amount if the microphysical changes affect precipitation and turbulent mixing.

**3.2 Robustness of single-year estimates**

A major limitation of the present analysis is the reliance on estimates derived from single years of two-month data. Even for the clear increase in Nd observable over the shipping corridor, there is substantial noise in single-year estimates of the shipping effect (e.g., Fig. 6 in Diamond et al., 2020). To test how unusual it would be to observe something like the decrease in Nd perturbation in all years after 2019 and before 2024, we repeat the analysis above for VIIRS on NOAA-2020 (available since 2018) with values from the MODerate resolution Imaging Spectrometer (MODIS) on Aqua (available since 2002, with increasing orbital drift since 2022; see Methods in Appendix A). Figure 3 shows how each individual year's data compares to the pre-2020 climatology, the 2020-2023 period, and 2024. Using a Monte-Carlo approach, we randomly draw 100,000 samples, with replacement, of four-year sets from the population of single-year (September-October) Aqua/MODIS data from 2003-2019 and test how frequently the mean Nd for all four years is less than half the climatological 2003-2019 value (as was true for 2020-2023). Such a situation occurs in less than 0.02% of cases. It is thus exceedingly unlikely that the pattern observed in the single-year MODIS or VIIRS data since IMO 2020 went into effect is due to chance.

Figure 3. Probability densities (via Gaussian kernel density estimation) of relative Nd perturbations for the pre-2020 September-October climatology (blue curve), 2020-2023 period (yellow curve), and 2024 (red curve), with 95% confidence intervals shown as transparent shading, and individual year mean estimates (colored markers) and their 95% confidence intervals (error bars) from Aqua/MODIS.

125

As a further check of the robustness of our results, we analyze whether any of the years between 2018 and 2024 experienced unusual meteorological or non-shipping aerosol conditions (Figure 4; see Methods in Appendix A). In particular, we are interested in assessing whether the clouds would likely have been drizzling in the absence of shipping aerosol; if 2024 were the only post-IMO 2020 years with likely background drizzle, that could be an alternate explanation for why the cloud microphysics were particularly susceptible to the remaining shipping perturbation that year. We assess the background cloud fraction (lower values would indicate drizzle) and cloud droplet effective radius ( $r_e$ ; higher for drizzle). Aerosol optical depth (AOD) would generally indicate drizzle if low (fewer CCN), but in this region is more complicated given that much of the AOD in this season is above-cloud smoke that does not activate to form cloud droplets (Diamond et al., 2018) but could bias

the cloud property retrievals (Haywood et al., 2004; Meyer et al., 2015). For background cloud fraction, re, and AOD we exclude all potentially shipping-affected values (see Methods in Appendix A). We also look at a selection of cloud-controlling factors with the expectation that lower sea surface temperatures (SST), greater estimated inversion strength (EIS), and greater wind speed would be associated with thicker stratocumulus clouds (Eastman et al., 2022; Klein and Hartmann, 1993; Scott et al., 2020; Wood and Bretherton, 2006) and thus more drizzle, although the true effect of each could be more complicated and the cloud controlling factors may also interact with each other. We are thus interested in identifying outliers in either direction as potentially problematic for our conclusions about the marine fuel effects.

(a) (b) 15.5 (c) 0.38 COVID-19 pre-IMO 2020 0.36 14.5 € 0.34 돌 14.0 O A 0.32 e 13.5 13.0 12.5 0.28 2018 2019 2020 2021 2022 2023 2024 2018 2019 2020 2021 2022 2023 2024 2018 2019 2020 2021 2022 2023 2024 (d) 21.8 (f) 7.4 8.00 21.6 7.2 7.75 Wind speed (m s-1) 7.0 7.50 ŝ 7.25 6.8 SH 7.00 6.75 21.0 6.2 20.8 2018 2019 2020 2021 2022 2023 2024 6.0 2018 2019 2020 2021 2022 2023 2024 2018 2019 2020 2021 2022 2023 2024

Figure 4. Background cloud and aerosol properties (a-c) and meteorological cloud-controlling factors (d-f) over the southeast 4. Atlantic during September-October of each year (red box in Fig. 1d). Markers indicate mean values and error bars two standard errors. For the cloud and aerosol values (a-c), potentially shipping-affected areas are excluded.

Neither 2019 nor 2024 feature particularly unusual values for any of the quantities analyzed (Fig. 4). The only striking outlier noted is the high EIS in 2023 (Fig. 4e), which along with its high values in  $r_e$  and wind speed, would suggest a stronger likelihood of drizzle in the background clouds. Contra our expectations above, however, 2023 has an unusually negative estimate for the shipping corridor  $N_d$  perturbation (Fig. 2b). We thus conclude that unusual meteorology in any single year is unlikely to have affected our results.

**4 Conclusions**

155 Changes in ship traffic stemming from armed conflict around the Red Sea came just a few years after ship emissions had already changed dramatically due to sulfur pollution regulations; this unique "experiment-within-an-experiment" setup allows us to better quantify how cloud responses to shipping have changed following IMO 2020. By comparing the shipping perturbation on NO2, which is not strongly affected by fuel composition, with the shipping effect on CCN (as indicated by  $N_d$ ), which is very sensitive to the fuel composition, we derive a measure of the cloud-altering properties of a given quantity of fuel burned. The spike in shipping activity in the southeast Atlantic in 2024 due to disruption in Suez Canal traffic results in raw  $N_d$  perturbations rebounding from being indistinguishable from zero post-IMO to being nearly as large as the pre-2020 values; normalizing by the anomalously large NO2 perturbation, however, reveals a 66% (25-88%) reduction in the cloudaltering ability of the post-2020 fuel, at least in stratocumulus conditions. If the constraint on cloud microphysical sensitivity to shipping emissions established here could be extrapolated globally, it may provide an emergent constraint on historical radiative forcing from aerosol-cloud interactions from global climate models.

**Appendix A: Methods**

NO2 data (TM5-MP-DOMINO v2.8.0) is from the TROPOspheric Monitoring Instrument (TROPOMI) on Sentinel 5-P (Copernicus Sentinel-5P, 2021; van Geffen et al., 2022). The shipping corridor contribution to column NO2 is calculated by first finding the longitude with the greatest shipping NOx emissions from the Emissions Database for Global Atmospheric Research (EDGAR; Crippa et al., 2020) for each latitude in the domain 13.5°W-3.5°E, 15°S-0° and compositing the data 5° east and west of these points. We then fit a line using simple ordinary least squares linear regression connecting the data between 2-5° to the east and west (Fig. A1a) and subtract the regressed values from the observed values within 0.25° east and west of the central point (Fig. A1b). Errors are quantified by propagating the sampling uncertainty in the observed mean value and the root mean square error of the regression.

Moved up [1]: Figure 3. Probability densities (via Gaussian kernel density estimation) of relative Nd perturbations for the pre-2020 September-October climatology (blue curve), 2020-2023 period (yellow curve), and 2024 (red curve), with 95% confidence intervals shown as transparent shading, and individual year mean estimates (colored markers) and their 95% confidence intervals (error bars) from Aqua/MODIS.¶

Figure A1. Example profiles from September-October 2024 of the method for estimating nitrogen dioxide shipping perturbations. After aggregating observations within 5° east and west of the shipping corridor (a, solid line), a background estimate is created using data from 2-5° on either side of the corridor (a, dashed line). The difference between these values (b) is interpreted as the perturbation caused by shipping activity. Shading represents 95% confidence based on sampling (a, blue) or the regression fit (a, red) or their propagated combination (b). Dotted lines represent the region within 0.25° of the shipping corridor shown in Figure 2.

 $N_{\rm d}$  is calculated (Grosvenor et al., 2018) from the Visible Infrared Imaging Radiometer Suite (VIIRS) on NOAA-20 and the MODerate resolution Imagaing Spectrometer (MODIS) on Aqua (Platnick et al., 2017) using joint histograms of liquid cloud droplet effective radius and optical thickness following Painemal and Zuidema (2011). The shipping corridor contribution to  $N_{\rm d}$  within a domain from 18-8°S and 13°W-8°E is estimated by generating a counterfactual field of non-shipping-influenced values following the universal kriging methodology of Diamond et al. (2020). The definition of potentially shipping-affected pixels follows Diamond (2023) but is expanded 2° to the west; the "core" corridor effect is quantified as the average difference between 5000 simulated plausible counterfactual values and the observed value at the grid points of the greatest shipping SO2 emissions (Crippa et al., 2020) and 2° to the west at each latitude, Sea surface temperature from the Extended Reconstructed Sea Surface Temperature, Version 5 dataset (Huang et al., 2017) is used as the cloud controlling factor in fitting the mean function.

200

The aerosol and cloud properties and cloud-controlling factors shown in Figure 4 are from the Clouds and the Earth's Radiant Energy System (CERES) Single-Scanner Footprint Edition 1C level 3 monthly product for NOAA-20 (NASA/LARC/SD/ASDC, 2020). Cloud fraction and cloud effective radius are retrieved using VIIRS radiances (Minnis et al.,

2023; Yost et al., 2023). AOD at 550 nm is retrieved using the Deep Blue algorithm (Hsu et al., 2013). SST, EIS, and wind speed data are from the Modern-Era Retrospective Analysis for Research and Applications, Version 2 (Gelaro et al., 2017).

[revised manuscript text omitted]

  Eastman, R., McCoy, I. L., and Wood, R.: Wind, Rain, and the Closed to Open Cell Transition in Subtropical Marine
- Estiman, K., McCoy, I. E., and Wood, K.: Whild, Rain, and the Closed to Open Cen Transition in Subtoplical Marine Stratocumulus, Journal of Geophysical Research: Atmospheres, 127, 10.1029/2022jd036795, 2022.

  England, M. H., Li, Z., Huguenin, M. F., Kiss, A. E., Sen Gupta, A., Holmes, R. M., and Rahmstorf, S.: Drivers of the extreme
- North Atlantic marine heatwave during 2023, Nature, 642, 636-643, 10.1038/s41586-025-08903-5, 2025. European Commission Joint Research Centre: Emissions Database for Global Atmospheric Research [Data set]. Joint Research Centre Data Catalogue, doi:10.2904/JRC DATASET EDGAR, 2018.
- Forster, P. M., T. Storelvmo, K. Armour, W. Collins, J.-L. Dufresne, D. Frame, D.J. Lunt, T. Mauritsen, M.D. Palmer, M.
- 290 Watanabe, M. Wild, and Zhang, H.: The Earth's Energy Budget, Climate Feedbacks, and Climate Sensitivity, in: Climate

- Change 2021: The Physical Science Basis. Contribution of Working Group I to the Sixth Assessment Report of the Intergovernmental Panel on Climate Change, Cambridge University Press, Cambridge, United Kingdom and New York, NY, USA, 923–1054, 2021.
- Gelaro, R., McCarty, W., Suárez, M. J., Todling, R., Molod, A., Takacs, L., Randles, C. A., Darmenov, A., Bosilovich, M. G.,
  Reichle, R., Wargan, K., Coy, L., Cullather, R., Draper, C., Akella, S., Buchard, V., Conaty, A., da Silva, A. M., Gu, W., Kim,
  G.-K., Koster, R., Lucchesi, R., Merkova, D., Nielsen, J. E., Partyka, G., Pawson, S., Putman, W., Rienecker, M., Schubert, S.
  D., Sienkiewicz, M., and Zhao, B.: The Modern-Era Retrospective Analysis for Research and Applications, Version 2 (MERRA-2), Journal of Climate, 30, 5419-5454, 10.1175/jcli-d-16-0758.1, 2017.
  Gettelman, A., Christensen, M. W., Diamond, M. S., Gryspeerdt, E., Manshausen, P., Stier, P., Watson-Parris, D., Yang, M.
- 300 Yoshioka, M., and Yuan, T.: Has Reducing Ship Emissions Brought Forward Global Warming?, Geophysical Research Letters, 51, 10.1029/2024g1109077, 2024.
  Grosvenor, D. P., Sourdeval, O., Zuidema, P., Ackerman, A., Alexandrov, M. D., Bennartz, R., Boers, R., Cairns, B., Chiu, J.
  - Grosvenor, D. P., Sourdeval, O., Zuidema, P., Ackerman, A., Alexandrov, M. D., Bennartz, R., Boers, R., Cairns, B., Chiu, J. C., Christensen, M., Deneke, H., Diamond, M., Feingold, G., Fridlind, A., Hünerbein, A., Knist, C., Kollias, P., Marshak, A., McCoy, D., Merk, D., Painemal, D., Rausch, J., Rosenfeld, D., Russchenberg, H., Seifert, P., Sinclair, K., Stier, P.,
- 305 van Diedenhoven, B., Wendisch, M., Werner, F., Wood, R., Zhang, Z., and Quaas, J.: Remote Sensing of Droplet Number Concentration in Warm Clouds: A Review of the Current State of Knowledge and Perspectives, Reviews of Geophysics, 56, 409–453, 10.1029/2017rg000593, 2018.
  Grysneerdt, F., Smith, T. W. P., Olfceffe, F., Christensen, M. W., and Goldsworth, F. W.: The Impact of Ship Emission
  - Gryspeerdt, E., Smith, T. W. P., O'Keeffe, E., Christensen, M. W., and Goldsworth, F. W.: The Impact of Ship Emission Controls Recorded by Cloud Properties, Geophysical Research Letters, 46, 12547–12555, 10.1029/2019gl084700, 2019.
- 310 Harris, C. R., Millman, K. J., van der Walt, S. J., Gommers, R., Virtanen, P., Cournapeau, D., Wieser, E., Taylor, J., Berg, S., Smith, N. J., Kern, R., Picus, M., Hoyer, S., van Kerkwijk, M. H., Brett, M., Haldane, A., Del Rio, J. F., Wiebe, M., Peterson, P., Gerard-Marchant, P., Sheppard, K., Reddy, T., Weckesser, W., Abbasi, H., Gohlke, C., and Oliphant, T. E.: Array programming with NumPy, Nature, 585, 357-362, 10.1038/s41586-020-2649-2, 2020.
- Haywood, J. M., Osborne, S. R., and Abel, S. J.: The effect of overlying absorbing aerosol layers on remote sensing retrievals of cloud effective radius and cloud optical depth, Quarterly Journal of the Royal Meteorological Society, 130, 779-800, 10.1256/qj.03.100, 2004.
  - Hoyer, S., and Hamman, J. J.: xarray: N-D labeled Arrays and Datasets in Python, Journal of Open Research Software, 5, 10.5334/jors.148, 2017.
- Hsu, N. C., Jeong, M.-J., Bettenhausen, C., Sayer, A. M., Hansell, R., Seftor, C. S., Huang, J., and Tsay, S.-C.: Enhanced Deep
   Blue aerosol retrieval algorithm: The second generation, Journal of Geophysical Research: Atmospheres, 118, 9296-9315, <a href="https://doi.org/10.1002/jgrd.50712">https://doi.org/10.1002/jgrd.50712</a>, 2013.
   History Property of Computer Science W. The Dependance of Skin Polluted Marine
  - Hu, S., Zhu, Y., Rosenfeld, D., Mao, F., Lu, X., Pan, Z., Zang, L., and Gong, W.: The Dependence of Ship-Polluted Marine Cloud Properties and Radiative Forcing on Background Drop Concentrations, Journal of Geophysical Research: Atmospheres, 126, e2020JD033852, 10.1029/2020jd033852, 2021.
- 325 Huang, B., Thorne, P. W., Banzon, V. F., Boyer, T., Chepurin, G., Lawrimore, J. H., Menne, M. J., Smith, T. M., Vose, R. S., and Zhang, H.-M.: NOAA Extended Reconstructed Sea Surface Temperature (ERSST), Version 5 [Data set]. NOAA National Centers for Environmental Information, doi:10.7289/V5T72FNM, 2017.
  - Hunter, J. D.: Matplotlib: A 2D Graphics Environment, Computing in Science & Engineering, 9, 90-95, 10.1109/MCSE.2007.55, 2007.
- 330 Jordan, G., and Henry, M.: IMO2020 Regulations Accelerate Global Warming by up to 3 Years in UKESM1, Earth's Future, 12, 10.1029/2024ef005011, 2024.
  - Klein, S. A., and Hartmann, D. L.: The Seasonal Cycle of Low Stratiform Clouds, Journal of Climate, 6, 1587-1606, 1993. Manshausen, P., Watson-Parris, D., Christensen, M. W., Jalkanen, J. P., and Stier, P.: Rapid saturation of cloud water adjustments to shipping emissions, Atmos. Chem. Phys., 23, 12545-12555, 10.5194/acp-23-12545-2023, 2023.
- 335 March, D., Metcalfe, K., Tintore, J., and Godley, B. J.: Tracking the global reduction of marine traffic during the COVID-19 pandemic, Nat Commun, 12, 2415, 10.1038/s41467-021-22423-6, 2021.
  Met Office: Cartopy: a cartographic python library with a matplotlib interface. Met Office (Ed.), Exeter, Devon, 2010-2015.
  Meyer, K., Platnick, S., and Zhang, Z.: Simultaneously inferring above-cloud absorbing aerosol optical thickness and underlying liquid phase cloud optical and microphysical properties using MODIS, Journal of Geophysical Research:
- 340 Atmospheres, 120, 5524-5547, 10.1002/2015jd023128, 2015.

- Minnis, P., Sun-Mack, S., Smith, W. L., Trepte, Q. Z., Hong, G., Chen, Y., Yost, C. R., Chang, F.-L., Smith, R. A., Heck, P. W., and Yang, P.: VIIRS Edition 1 Cloud Properties for CERES, Part 1: Algorithm Adjustments and Results, Remote Sensing, 15, 578, 2023.
- NASA LAADS DAAC: CLDPROP\_M3\_MODIS\_Aqua MODIS/Aqua Cloud Properties Level 3 monthly, 1x1 degree grid.

  NASA Level-1 and Atmosphere Archive & Distribution System, doi:10.5067/MODIS/CLDPROP\_M3\_MODIS\_Aqua.011.

  NASA VIIRS Atmosphere SIPS: NOAA20 VIIRS Cloud Properties Level 3 monthly, 1x1 degree grid (CLDPROP\_M3\_VIIRS\_NOAA20). Space Science & Engineering Center, U. o. W.-M. (Ed.), NASA Level-1 and Atmosphere Archive & Distribution System, doi:10.5067/VIIRS/CLDPROP M3\_VIIRS\_NOAA20.011.
- NASA/LARC/SD/ASDC: CERES Time-Interpolated TOA Fluxes, Clouds and Aerosols Monthly NOAA-20 Edition1B.

  NASA Langley Atmospheric Science Data Center DAAC, 2020.
- Painemal, D., and Zuidema, P.: Assessment of MODIS cloud effective radius and optical thickness retrievals over the Southeast Pacific with VOCALS-REx in situ measurements, Journal of Geophysical Research: Atmospheres, 116, D24206, 10.1029/2011jd016155, 2011.

  Petzold, A., Weingartner, E., Hasselbach, J., Lauer, P., Kurok, C., and Fleischer, F.: Physical Properties, Chemical
- 355 Composition, and Cloud Forming Potential of Particulate Emissions from a Marine Diesel Engine at Various Load Conditions, Environmental Science & Technology, 44, 3800-3805, 10.1021/es903681z, 2010.
  Platnick, S., Meyer, K. G., King, M. D., Wind, G., Amarasinghe, N., Marchant, B., Arnold, G. T., Zhang, Z., Hubanks, P. A., Holz, R. E., Yang, P., Ridgway, W. L., and Riedi, J.: The MODIS cloud optical and microphysical products: Collection 6 updates and examples from Terra and Aqua, IEEE Transactions on Geoscience and Remote Sensing. 55, 502-525,
  - 10.1109/TGRS.2016.2610522, 2017.

    Pseftogkas, A., Stavrakou, T., Müller, J. F., Koukouli, M. E., Balis, D., and Meleti, C.: Shifts in Maritime Trade Routes as a Result of Red Sea Shipping Crisis Detected in TROPOMI NO2 Data, Geophysical Research Letters, 51, 10.1029/2024g1110491, 2024.
  - Quaglia, I., and Visioni, D.: Modeling 2020 regulatory changes in international shipping emissions helps explain anomalous 2023 warming, Earth Syst. Dynam., 15, 1527-1541, 10.5194/esd-15-1527-2024, 2024.
- Radke, L. F., Coakley, J. A., and King, M. D.: Direct and Remote Sensing Observations of the Effects of Ships on Clouds, Science, 246, 1146-1149, 1989.
  - Raghuraman, S. P., Soden, B., Clement, A., Vecchi, G., Menemenlis, S., and Yang, W.: The 2023 global warming spike was driven by the El Niño-Southern Oscillation, Atmos. Chem. Phys., 24, 11275-11283, 10.5194/acp-24-11275-2024, 2024.
- 370 Raydan, N., and Nadimi, F.: Tracking Maritime Attacks in the Middle East Since 2019, The Washington Institute for Near East Studies, 2025.
  geoR: Analysis of Geostatistical Data. R package version 1.7-5.2.1: <a href="https://CRAN.R-project.org/package=geoR">https://CRAN.R-project.org/package=geoR</a>, 2018.
  Scott, R. C., Myers, T. A., Norris, J. R., Zelinka, M. D., Klein, S. A., Sun, M., and Doelling, D. R.: Observed Sensitivity of Levi Cloud Rediction Effects to Meteorolegical Porturbations over the Cloud Cooper Levinge of Climbal 2023. 7711, 7724.
- Low-Cloud Radiative Effects to Meteorological Perturbations over the Global Oceans, Journal of Climate, 33, 7717-7734, 375 10.1175/jcli-d-19-1028.1, 2020.
- Song, H.: Datetimes and Locations of ship-tracks [Data set]. Harvard Dataverse, 2022.
  Toll, V., Christensen, M., Quaas, J., and Bellouin, N.: Weak average liquid-cloud-water response to anthropogenic aerosols, Nature, 572, 51-55, 10.1038/s41586-019-1423-9, 2019.
- Twomey, S.: The Influence of Pollution on the Shortwave Albedo of Clouds, Journal of the Atmospheric Sciences, 34, 1149-30 1152, 10.1175/1520-0469(1977)034<1149:tiopot>2.0.co;2, 1977.
- 580 1132, 10.11/5/1520-0469(197/)034
  1149:ftopot
  2.0.co;2, 197/.
  van Geffen, J., Eskes, H., Compernolle, S., Pinardi, G., Verhoelst, T., Lambert, J. C., Sneep, M., ter Linden, M., Ludewig, A., Boersma, K. F., and Veefkind, J. P.: Sentinel-5P TROPOMI NO2 retrieval: impact of version v2.2 improvements and comparisons with OMI and ground-based data, Atmos. Meas. Tech., 15, 2037-2060, 10.5194/amt-15-2037-2022, 2022.
- Veefkind, J. P., Aben, I., McMullan, K., Förster, H., de Vries, J., Otter, G., Claas, J., Eskes, H. J., de Haan, J. F., Kleipool, Q., van Weele, M., Hasekamp, O., Hoogeveen, R., Landgraf, J., Snel, R., Tol, P., Ingmann, P., Voors, R., Kruizinga, B., Vink, R., Visser, H., and Levelt, P. F.: TROPOMI on the ESA Sentinel-5 Precursor: A GMES mission for global observations of the atmospheric composition for climate, air quality and ozone layer applications, Remote Sensing of Environment, 120, 70-83, 10.1016/j.rse.2011.09.027, 2012.
- Virtanen, P., Gommers, R., Oliphant, T. E., Haberland, M., Reddy, T., Cournapeau, D., Burovski, E., Peterson, P., Weckesser, 390 W., Bright, J., van der Walt, S. J., Brett, M., Wilson, J., Millman, K. J., Mayorov, N., Nelson, A. R. J., Jones, E., Kern, R.,

- Larson, E., Carey, C. J., Polat, I., Feng, Y., Moore, E. W., VanderPlas, J., Laxalde, D., Perktold, J., Cimrman, R., Henriksen, I., Quintero, E. A., Harris, C. R., Archibald, A. M., Ribeiro, A. H., Pedregosa, F., van Mulbregt, P., and SciPy, C.: SciPy 1.0: fundamental algorithms for scientific computing in Python, Nat Methods, 17, 261-272, 10.1038/s41592-019-0686-2, 2020. Watson-Parris, D., Christensen, M. W., Laurenson, A., Clewley, D., Gryspeerdt, E., and Stier, P.: Shipping regulations lead to
- 395 large reduction in cloud perturbations, Proc Natl Acad Sci U S A, 119, e2206885119, 10.1073/pnas.2206885119, 2022. Watson-Parris, D., Wilcox, L. J., Stjern, C. W., Allen, R. J., Persad, G., Bollasina, M. A., Ekman, A. M. L., Iles, C. E., Joshi, M., Lund, M. T., McCoy, D., Westervelt, D. M., Williams, A. I. L., and Samset, B. H.: Surface temperature effects of recent reductions in shipping SO2 emissions are within internal variability, Atmos. Chem. Phys., 25, 4443-4454, 10.5194/acp-25-4443-2025, 2025.
- 400 Wood, R., and Bretherton, C. S.: On the Relationship between Stratiform Low Cloud Cover and Lower-Tropospheric Stability, Journal of Climate, 19, 6425-6432, 2006.
  - Yost, C. R., Minnis, P., Sun-Mack, S., Smith, W. L., and Trepte, Q. Z.: VIIRS Edition 1 Cloud Properties for CERES, Part 2: Evaluation with CALIPSO, Remote Sensing, 15, 1349, 2023.
- Yuan, T., Song, H., Wood, R., Wang, C., Oreopoulos, L., Platnick, S. E., von Hippel, S., Meyer, K., Light, S., and Wilcox, E.:
  405 Global reduction in ship-tracks from sulfur regulations for shipping fuel, Science Advances, 8, eabn7988, doi:10.1126/sciadv.abn7988, 2022.
  - Yuan, T., Song, H., Wood, R., Oreopoulos, L., Platnick, S., Wang, C., Yu, H., Meyer, K., and Wilcox, E.: Observational evidence of strong forcing from aerosol effect on low cloud coverage, Science Advances, 9, eadh7716, doi:10.1126/sciadv.adh7716, 2023.